# Intraneuronal binding of amyloid beta with reelin—Implications for the onset of Alzheimer's disease

**Asgeir Kobro-Flatmoen**[1,2]*, **Stig W. Omholt**[3]

**1** Kavli Institute for Systems Neuroscience, Norwegian University of Science and Technology (NTNU), Trondheim, Norway, **2** K. G. Jebsen Centre for Alzheimer's Disease, Norwegian University of Science and Technology (NTNU), Trondheim, Norway, **3** Department of Circulation and Medical Imaging, Norwegian University of Science and Technology (NTNU), Trondheim, Norway

* asgeir.kobro-flatmoen@ntnu.no

**Data Availability Statement:** All data are in the manuscript. Instead of depositing the original Python code, we have deposited a CellML version

## Abstract

Numerous studies of the human brain supported by experimental results from rodent and cell models point to a central role for intracellular amyloid beta (Aβ) in the onset of Alzheimer's disease (AD). In a rat model used to study AD, it was recently shown that in layer II neurons of the anteriolateral entorhinal cortex expressing high levels of the glycoprotein reelin (Re⁺alECLII neurons), reelin and Aβ engage in a direct protein–protein interaction. If reelin functions as a sink for intracellular Aβ and if the binding to reelin makes Aβ physiologically inert, it implies that reelin can prevent the neuron from being exposed to the harmful effects typically associated with increased levels of oligomeric Aβ. Considering that reelin expression is extraordinarily high in Re⁺alECLII neurons compared to most other cortical neurons, such a protective role appears to be very difficult to reconcile with the fact that this subset of ECLII neurons is clearly a major cradle for the onset of AD. Here, we show that this conundrum can be resolved if Re⁺alECLII neurons have a higher maximum production capacity of Aβ than neurons expressing low levels of reelin, and we provide a rationale for why this difference has evolved.

## Author summary

Amyloid-β is a small peptide that is widely recognized as one of the main culprits involved in the development of Alzheimer's disease. It was recently shown that in a subset of neurons in layer II of a brain area called the entorhinal cortex that expresses extraordinarily high levels of the protein reelin, amyloid-β and reelin bind to each other. These neurons, which are strongly involved in memory formation, are among the first to die in subjects with Alzheimer's disease. If intracellular amyloid-β becomes physiologically inert when it binds to reelin, it suggests that reelin can protect these neurons from being exposed to the detrimental effects of increased levels of amyloid-β. Such an effect seems very difficult to reconcile with the fact that these neurons constitute the predominant cortical site for the initiation of Alzheimer's disease. Here, we use a mathematical model to show that this

of the model in the Physiome Model Repository available at: https://models.physiomeproject.org/workspace/be4. We chose this option because the purpose of this repository is both to serve as a permanent store and to allow scientists to share models even if they are using different modeling tools. In addition, a CellML model is curated to ensure consistent use of dimensions and that the provided model and associated parameter values do indeed produce the reported results.

**Funding:** This work was supported by the Center Grant from (1) The Liaison Committee for Education, Research and Innovation in Central Norway (https://www.helse-midt.no/samarbeidsorganet), (2) The K. G. Jebsen Foundation (http://www.kgjf.org/), The (3) The Kavli Foundation (https://www.kavlifoundation.org/), and (4) and Individual grant from Civitan (https://civitan.no/alzheimerfondet/informasjon-om-fondet/) to AKF. The funders had no role in study design, data collection and analysis, decision to publish, or preparation of the manuscript.

**Competing interests:** The authors have declared that no competing interests exist.

paradox may be resolved if we allow neurons expressing high levels of reelin to have a higher maximum capacity to produce amyloid-β.

## Introduction

Alzheimer's disease (AD) is a neurodegenerative brain disease that causes dementia, and the small peptide amyloid-$\beta$ (Aβ) and hyperphosphorylated versions of the tau protein (p-tau) have been identified as central in the development of the disease. Studies of large human cohorts strongly suggest that the disease begins with an increase in Aβ in the brain, leading to the generation of p-tau and eventually the formation of insoluble p-tau aggregates, called neurofibrillary tangles (NFTs) [1, 2].

The vast majority of AD subjects experience progressive impairment of declarative memory, whose function depends on the medial temporal lobe memory system. A pivotal component of this system is the entorhinal cortex (EC), which constitutes the main gateway for information flow between the neocortex and the hippocampus and plays a crucial role in the formation, consolidation, and retrieval of memories and spatiotemporal representations. Long before clinically recognized symptoms of AD appear, such as substantial or persistent memory loss, EC is already severely degenerated [3, 4], most notably in its anteriolateral portion (alEC) [5, 6]. Discovering the mechanisms responsible for this impairment is arguably of considerable importance if we want to achieve a truly efficient therapy against AD.

The function of the large glycoprotein reelin in adult mammalian brains includes synaptic modulation, induction of enhanced spine density, and promotion of long-term potentiation [7]. Most neurons in layer II of alEC are unique among cortical excitatory neurons by expressing extraordinarily high levels of reelin (dubbed Re⁺alECLII neurons in the following). In terms of existing data and interpretations of AD development, at least three possibly interlinked paradoxes are attached to this observation.

Numerous studies of the human brain using live imaging, immunohistochemistry, and biochemistry, supported by experimental results from rodent and cell models, point to a role for intracellular Aβ in non-fibrillated forms in the onset of Alzheimer's disease [1]. The first paradox arises from the recent observation that in Re⁺alECLII neurons in wild type rats as well as in McGill-R-Thy1-APP homozygous transgenic rats (a model commonly used to study AD), reelin and intracellular Aβ engage in a direct protein–protein interaction [8]. This is an intriguing discovery, as it suggests that reelin can function as a sink for intracellular Aβ. If we assume that the Aβ-reelin complex is physiologically inert, this would imply that ECLII neurons with a high reelin level would be better protected from the detrimental effects associated with increased intracellular Aβ expression. However, this does not resonate with the observation that these neurons are among the first to die when the AD phenotype unfolds.

The second paradox arises from our current knowledge of reelin signaling and p-tau formation. Reelin is probably important for the formation of declarative memories, and when it binds to its main receptor in the brain, the apolipoprotein E receptor 2 (ApoER2), it triggers a signaling cascade that increases glutamatergic transmission [7]. Additionally, the binding of reelin to ApoER2 activates a signaling cascade that strongly inhibits the activity of glycogen synthase kinase 3β (GSK3β) [9–11]. GSK3β is one of the main kinases that phosphorylates tau, and constitutive tau phosphorylation causes NFT formation. It has been experimentally demonstrated that interaction with Aβ reduces the capacity of reelin to inhibit GSK3β kinase activity [12]. This suggests that a high intracellular reelin level would buffer against NFT formation in a senescent physiology that causes frequent bouts of increased expression of Aβ. However,

this does not appear to agree with the observation that Re[+]ECLII projection neurons that reside superficially in layer II are typically the first to form NFTs [2, 13, 14].

The third paradox arises from the relationship between the reelin function and the frequently advocated hypothesis that infectious agents contribute to the pathogenesis of AD. That infections can cause AD has been a controversial issue for decades. However, in the last decade, a substantial body of data has accumulated that at least supports the notion that Aβ has antiviral [15, 16] and antimicrobial properties [17, 18]. And the data support that Aβ oligomers are capable of binding to viruses [16, 19] and microbes [20]. Furthermore, the antimicrobial properties of Aβ have been reported to be similar to the canonical human antimicrobial peptide LL-37, showing an antimicrobial activity equivalent to or, in some cases, superior to this cathelicidin [21] against various clinically relevant bacteria [17]. Therefore, several researchers have proposed that Aβ participates in an immune response to acute microbial or viral infection of the brain [22–25]. Assuming that such an immune response would trigger an increase in the expression of Aβ during pathogen infection, it follows that in Re[+]alECLII neurons Aβ would be prevented from exerting its immunological function by becoming attached to reelin. Although actual transport mechanisms and subsequent spread of viral infections through the central nervous system (CNS) are still poorly understood [26], the available data clearly suggest that ECLII neurons can be exposed to viral infection through their connection to olfactory receptor neurons and serve as a portal for further CNS infection [26–28]. Furthermore, the EC receives vascular input from the posterior and middle cerebral arteries, which forms particularly dense reticulated networks around individual clusters of ECLII neurons. Together with the neuronal clusters themselves, these networks give rise to characteristic bumps on the surface of the EC, known as the entorhinal verrucae. This redundancy in vascular input suggests that EC may also be particularly exposed to bloodborne pathogens [29]. Due to this propensity to become exposed to pathogens and assuming that increased Aβ production is indeed part of an intraneuronal immune response under native physiological conditions, it does not make evolutionary sense that Re[+]ECLII neurons should be more vulnerable to pathogen infection than other neurons.

Anticipating that resolving the third paradox may provide clues to resolving the other two, we constructed a simple mathematical model that describes the dynamics between intracellular Aβ, reelin and p-tau formation through the GSK3β pathway as a function of short-term pathogen exposure. We found that if the maximum possible production rate of Aβ is markedly higher in Re[+]alECLII neurons than in other cortical neurons, Re[+]alECLII neurons are capable of invoking an immune response similar to that of cortical neurons with low constitutive levels of reelin. This implies that in a senescent physiology predisposing to frequent inflammation-driven Aβ production bursts, also Re[+]alECLII neurons will be exposed to the detrimental effects of high levels of intracellular Aβ and increased p-tau formation, which resolves the other two paradoxes. Furthermore, we validate the model by showing that it is capable of predicting phenotypic effects associated with two genotypes on opposite sides of the AD risk spectrum, and by showing that it can accurately predict recent experimental data obtained by lowering the level of reelin in Re[+]alECLII neurons in McGill-R-Thy1-APP rats.

## Materials and methods

### Description of the dynamic model

The model consists of four differential equations that describe the time rate of change of the number of free intracellular Aβ42 molecules ([$A\beta$]) (the most prominent AD-associated type of Aβ), the number of free intracellular reelin molecules ([$Reelin$]), the number of intracellular Aβ molecules bound to reelin ([$A\beta_{reelin}$]), and the number of phosphorylated GSK3β molecules

being inhibited from inducing hyperphosphorylated tau production ($[GSK3\beta_p]$). The model is used to study the dynamics of these state variables in Re$^+$alECLII neurons and in cortical neurons that express low levels of reelin (referred to as LR neurons below). The four differential equations are the following.

$$\frac{d[A\beta]}{dt} = \alpha(t) - \beta[A\beta] - \gamma[A\beta][Reelin], \tag{1}$$

$$\frac{d[Reelin]}{dt} = \tau - \rho[Reelin] - \gamma[A\beta][Reelin], \tag{2}$$

$$\frac{d[A\beta_{reelin}]}{dt} = \gamma[A\beta][Reelin] - \delta[A\beta_{reelin}], \tag{3}$$

$$\frac{d[GSK3\beta_p]}{dt} = \eta[Reelin]\left(1 - \frac{[GSK3\beta_p]}{[GSK3\beta_{tot}]}\right) - \kappa[GSK3\beta_p]. \tag{4}$$

If we let $\alpha(t)$ be a constant $\alpha$, the steady state solution of this system can be expressed in closed form as:

$$[A\beta]^\star = \frac{\alpha\gamma - \beta\rho - \gamma\tau + Q}{2\beta\gamma}, \tag{5}$$

$$[Reelin]^\star = \frac{-\alpha\gamma - \beta\rho + \gamma\tau + Q}{2\gamma\rho}, \tag{6}$$

$$[A\beta_{reelin}]^\star = \frac{\alpha\gamma + \beta\rho + \gamma\tau - Q}{2\delta\gamma}, \tag{7}$$

$$[GSK3\beta_p]^\star = \frac{G_{tot}^2\eta\kappa Q - G_{tot}\eta(G_{tot}\alpha\gamma\kappa + G_{tot}\beta\kappa\rho - G_{tot}\gamma\kappa\tau + 2\beta\eta\tau)}{2G_{tot}^2\gamma\kappa^2\rho - 2G_{tot}\alpha\eta\gamma\kappa - 2G_{tot}\beta\eta\kappa\rho + 2G_{tot}\eta\gamma\kappa\tau - 2\beta\eta^2\tau}, \tag{8}$$

where $Q = \sqrt{\alpha^2\gamma^2 + 2\alpha\beta\gamma\rho - 2\alpha\gamma^2\tau + \beta^2\rho^2 + 2\beta\gamma\rho\tau + \gamma^2\tau^2}$ and $G_{tot} = [GSK3\beta_{tot}]$.

To be precise, the system has an additional steady state solution. But with the parameterization given below, this steady state does not need to be considered, as in this case three of the four state variables have negative values.

## Explanation of the differential equations

Eq (1) says that the rate of change of free Aβ is a function of the rate of production of Aβ at time $t$ ($\alpha(t)$), a first-order decay of Aβ ($\beta$ = constant) and a second-order binding reaction between Aβ and reelin ($\gamma$ = constant). We assume a 1:1 binding of reelin to Aβ. Here and in Eqs (2) and (3), the decay term denotes the rate at which the molecular species is tagged for removal. We are agnostic about whether this removal is done by means of the ubiquitin proteasome system, the endosome-autophagosome-lysosome system, the endoplasmic reticulum/Golgi-dependent secretory pathway or by some other process.

Eq (2) says that the rate of change in free reelin is a function of the constitutive production rate of reelin, a first-order decay of free reelin ($\rho$ = constant), and a second-order binding reaction between Aβ and reelin (identical to the last term in Eq (1)). Since in this study we compare the dynamics in two categories of neurons, LR neurons and Re$^+$alECLII neurons, it rests on the assumption that reelin is present in the vast majority of cortical neurons. To confirm that

this is indeed the case, we performed a quantitative analysis of brain image data from 3, 12 and 18-month-old wild type rats. The analysis showed that almost all of the about 172500 neurons analyzed, located in five different cortical regions, possess a reelin signal that is clearly above the background signal (Fig 1). Due to the high intracellular concentration, it is very likely that reelin is produced in Re$^+$alECLII neurons. However, we do not know whether this is the case in neurons with low constitutive levels of reelin. It may be that the reelin in these cells is delivered by interneurons [30]. If so, the parameter $\tau$ will have to be viewed as an import term. However, this does not affect the model as long as low-reelin neurons are provided with about the same amount of reelin per time unit.

Eq (3) says that the rate of change of the amount of the A$\beta_{reelin}$ complex is a function of its production rate (identical to the last term in Eq (1)) and first-order decay of the complex ($\delta$ = constant). In this paper, we do not account for the possibility that A$\beta_{reelin}$ complexes aggregate into configurations that may make them less exposed to intracellular degradation or export, which warrants a more complex decay term. Since here we are focused on describing a short-term immune response in native physiology, we can arguably assume that A$\beta_{reelin}$ under this condition is physiologically inert and does not have a notable impact on the formation rates of A$\beta$ or p-tau.

Eq (4) says that the rate of change in the number of phosphorylated GSK3$\beta$ molecules prevented from inducing tau hyperphosphorylation is given by the rate of GSK3$\beta$ phosphorylation due to reelin signaling minus a first-order rate process by which phosphorylated GSK3$\beta$ becomes dephosphorylated ($\kappa$ = constant). The parameter $\eta$ is constant and [$GSK3\beta_{tot}$] is the constitutive number of GSK3$\beta$ molecules that must be phosphorylated to completely inactivate the GSK3$\beta$ pathway. The term $(1 - [GSK3\beta_p]/[GSK3\beta_{tot}])$ captures that the phosphorylation rate will decrease with a decrease in the amount of unphosphorylated GSK3$\beta$. Reelin inactivates GSK3$\beta$ by binding to ApoER2 as a dimer [31]. Binding induces a receptor cycle process that activates a signaling cascade that eventually causes the phosphorylation of GSK3$\beta$ [26], inhibiting its kinase activity. Above a certain level of free reelin, the receptor cycling rate is likely to saturate. It is reason to believe that the functional form of this process is more sigmoidal than hyperbolic because reelin binds to ApoER2 as a dimer and because chemical reactions where one of the reactants is bound to a surface cause so-called fractal kinetics where even reactions involving no cooperativity will behave as they did [32–34]. However, due to the resolution level at which we model, we have not incorporated this mechanism since the term $\eta(1 - [GSK3\beta_p]/[GSK3\beta_{tot}])$ is sufficient to achieve the almost switchlike relationship between $[GSK3\beta_p]^\star$ and the reelin concentration that we assume to exist. The rationale being that with such a mechanism, GSK3$\beta$ remains close to fully phosphorylated beyond a modest free reelin concentration, thus preventing the formation of p-tau. Tau is a signaling hub protein that interacts with a wide variety of kinases and phosphatases, including the Src family kinases cSrc, Fgr, Fyn, and Lck [35]. For example, tau phosphorylation increases its affinity for the tyrosine kinase Fyn [36]. Hence, most likely, p-tau has several functions in native physiology whose orchestration needs to be deliberately regulated. And a switchlike mechanism ensures that the GSK3$\beta$ pathway is inactivated until its induction is needed, signalled by a substantial reduction in the level of free reelin.

Based on the Fig 1 data, we assume that the two types of neurons we study, Re$^+$alECLII neurons and LR neurons, have identical GSK3$\beta$ regulation such that the functional relationships that determine the propagation of the signal between reelin and GSK3$\beta$ are the same. That is, both groups of neurons produce (or receive) sufficient amounts of reelin to suppress the activation of p-tau formation through GSK3$\beta$ under native physiological conditions, and this suppression begins to fail when the intracellular level of free reelin falls below a certain threshold.

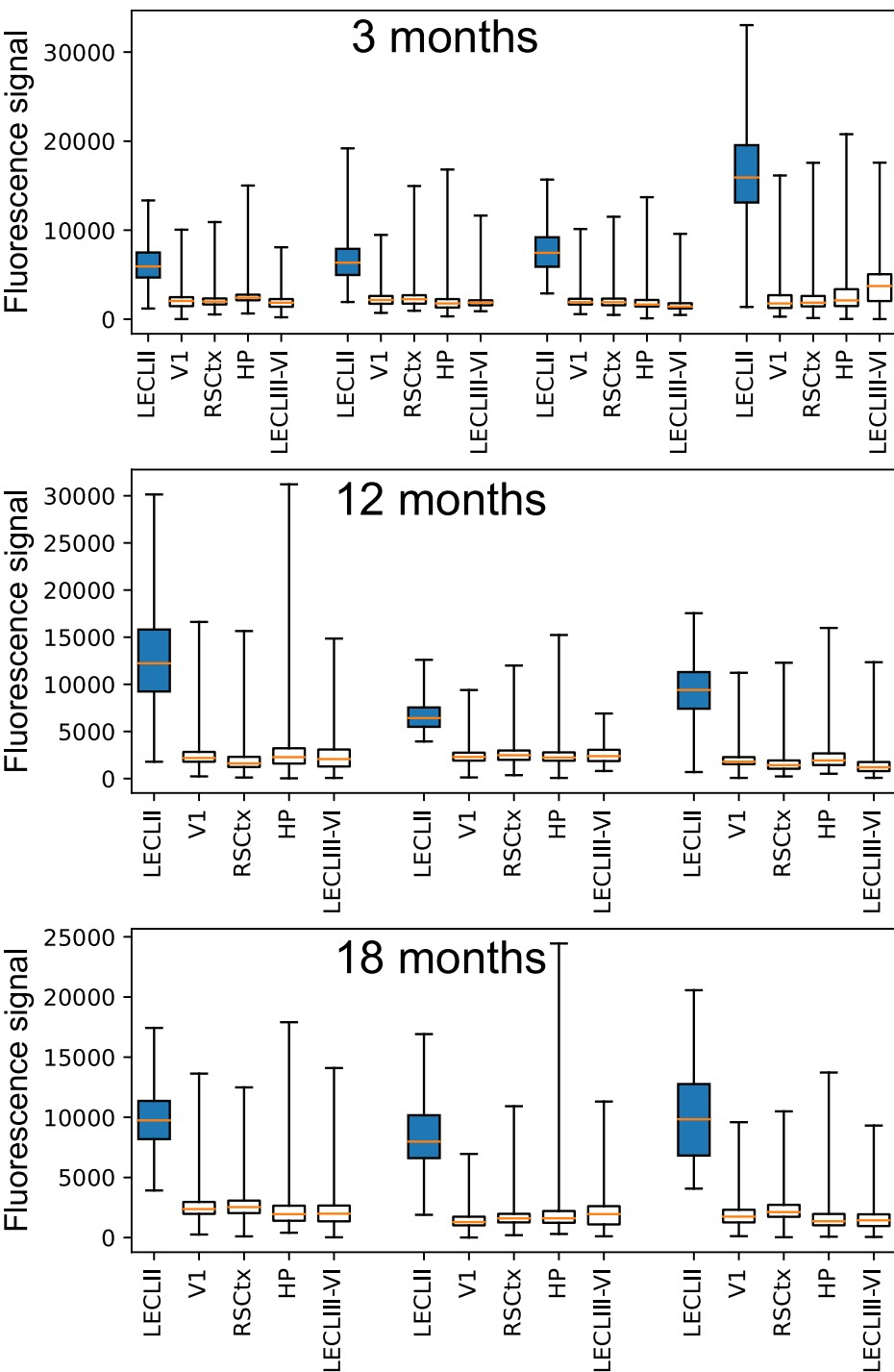

**Fig 1. Quantification of reelin levels across cortical regions in wild type rats.** Ten animals from three age groups were used in the study (3 months: 4; 12 months: 3; 18 months: 3). Brains were processed as detailed in reference [8]. Briefly, we quantified reelin-levels by way of densitometric measurement of cell profiles (primary antibody = Mouse anti-reelin (G10), Merk, Cat# MAB5364; RRID: AB_2179313; secondary antibody = Alexa 635 Goat anti-mouse (ThermoFisher, Cat# A-31574) in QuPath (Version 0.5.0). For each animal, we randomly selected three sections between Bregma level -5.64 and -6.48 on which the quantifications were done. On these sections, we selected our regions of interest (ROI) to cover the full extent of the cell layers. We adjusted the settings to include the maximum number of detections, such that for each ROI, virtually all available space was separated into unique cell profiles. Then, all cell profiles were assayed. For each section, we measured the signal in the underlying cortical white matter and used this for background subtraction. In rare instances, cell profiles where detected at borders where the tissue was very thin

due to tearing. Occasionally, following background subtraction, these instances gave rise to negative reelin signals (0.02% of the total cell profiles). These cases were deleted from the dataset. In all boxplots, the whiskers cover the whole range of data. LECLII = anteriolateral entorhinal cortex layer II (marked in blue), V1 = primary visual cortex, RSCtx = retrosplenial cortex, HP = hippocampus, LECLIII-VI = anteriolateral entorhinal cortex layers III-VI.

## The feedback structure of the model

The Jacobian $\mathbb{J}_{i,j} = \dfrac{\partial f_i(\mathbf{x})}{\partial x_j}$ of the system is:

$$
\mathbb{J}_{i,j} =
\begin{bmatrix}
-\beta - \gamma x_2 & -\gamma x_1 & 0 & 0 \\
-\gamma x_2 & -\gamma x_1 - \rho & 0 & 0 \\
\gamma x_2 & \gamma x_1 & -\delta & 0 \\
0 & \eta\left(1 - \frac{x_1}{G_{tot}}\right) & 0 & -\kappa - \frac{\eta x_2}{G_{tot}}
\end{bmatrix},
$$

where $x_1 = [A\beta]$, $x_2 = [Reelin]$, and $x_4 = [GSK3\beta_p]$.

We see from the Jacobian [37] that the system contains four negative autoregulatory loops (the diagonal elements) and one positive feedback loop between the time rate of change of $[A\beta]$ and the time rate of change of $[Reelin]$, i.e. the sign of element (1,2) x element (2,1) is positive. We have found it challenging to further simplify the model without losing essential features.

## Linear stability analysis

The Jacobian matrix was also used to study the stability of the baseline steady states of Re$^+$alE-CLII neurons and LR neurons given by (Eqs (5)–(8)), using the parameter sets specific to each neuron type (see below). More specifically, we made the above symbolic Jacobian matrix numerical by exchanging the parameter symbols with the numerical parameter values specific for each type of neuron and exchanging the variables $x_1$, $x_2$, and $x_4$ with the baseline steady state values of [Aβ], [Reelin], and [GSK3β$_p$] specific for each type of neuron (Eqs (5), (6) and (8)). We then calculated the eigenvalues of the two numerical matrices. Since all four eigenvalues in both cases were found to be negative real numbers, the two baseline steady states are locally asymptotically stable. This is also the case for the two steady states obtained when using a high $\alpha$ value to mimic infection, but we do not make use of these steady state values.

## Parameterization of the model

**Estimation of soma volumes.**   To keep things simple, we assume the dynamics described by Eqs (1)–(4) is confined to the soma. Given the soma volume, this allows us to translate molecular numbers into molar concentrations. Since different immunohistological methods or subtle differences in the use of the same method can affect the degree of shrinkage of the tissue under investigation, one must be cautious when attempting to compare the size of neuronal somata between different regions of the brain based on published work. With this in mind, we scrutinized the literature to obtain approximate measures of the size of neuronal somata in Re$^+$alECLII neurons versus LR neurons located in two areas where amyloid plaques tend to appear early in AD, namely the orbitofrontal cortex and the posterior cingulate cortex. In humans, available data [38, 39] suggest that the biggest Re$^+$alECLII neurons may have somata volumes greater than 12000 $\mu m^3$, but often have volumes much lower than this. The largest pyramidal neurons found in the posterior cingulate cortex [40] appear to have somata volumes

greater than 6000 $\mu m^3$. For the orbitofrontal cortex, most available data indicate that the volumes of the largest neuronal somata are somewhat lower than this [41, 42], although there are indications that more voluminous somata could be present here [43]. Although alECLII contains neurons with very large somata, they are still not the largest in the cortex, as the somata of pyramidal neurons in layer V can reach volumes many times greater than 12000 $\mu m^3$ [44]. We do not know whether the first AD-related changes develop in $Re^+$alECLII neurons with large or moderate somata volumes. And since the range of soma volume of these neurons overlaps with those of the orbitofrontal cortex and the posterior cingulate cortex, for simplicity, we assumed that the soma volumes of the two neuron groups under study were approximately 4850 $\mu m^3$.

**Eq (1) parameters.** The rate of neuronal secretion of Aβ has been estimated to be in the range of 2-4 $molecules \cdot s^{-1}$ [45], and the ratio of Aβ42 to Aβ40 in the cerebrospinal fluid of young human subjects has been estimated to be approximately 0.15 [46]. Assuming that these figures reflect the baseline levels, we let $\alpha_{baseline}$ = 1560 $molecules \cdot h^{-1}$ in LR neurons. In healthy Wistar rats, a low level of intracellular Aβ42 is found in neurons throughout the brain at all ages. However, in alECLII neurons, in particular those located in the superficial part of this layer, which is the part that contains the $Re^+$alECLII neurons, and in hippocampal neurons at the CA1/subiculum border [47], the Aβ42 signal is markedly stronger. We do not have quantitative measurements, but the signal is likely 4-5 times stronger. Due to the phylogenetic conservation of the EC, this pattern is likely to also be present in humans. This notion is supported by available data [47–50]. In accordance with this, we let $\alpha_{baseline}$ = 7020 $molecules \cdot h^{-1}$ in $Re^+$alECLII neurons (i.e. 4.5 times that of LR neurons).

We let the half-life of free Aβ in both types of neurons be 6.9 hours, which is 75% of the median half-life of intracellular proteins in mammalian cells [51], giving $\beta$ = 0.1 $h^{-1}$, and we let the binding rate between reelin and Aβ be $\gamma$ = 0.005 $molecules^{-1} \cdot h^{-1}$. The results are quite insensitive to the value of $\gamma$.

**Eq (2) parameters.** Since most surface glycoproteins have a half-life of 10-20 hours, and those with protein binding have a median half-life of approximately 19.5 hours [52], we let the free reelin half-life to be 15 hours, giving $\rho$ = 0.05 $h^{-1}$. Considering the median concentration of signal proteins in the brain is estimated to be 0.087 $\mu M$ [53], we assumed the free reelin concentration in LR neurons under baseline conditions to be approximately 0.088 $\mu M$. With the given values of $\alpha_{baseline}$, $\beta$, $\gamma$ and $\rho$, it follows from Eqs (5) and (6) that the estimated constitutive production rate (or import rate) of reelin in LR neurons is $\tau$ = 14500 $molecules \cdot h^{-1}$. Previous experience with immunohistochemical reelin staining [8, 54] indicates that the strong reelin signal obtained from $Re^+$alECLII neurons reflects a reelin concentration that is at least 5-6 times higher than what is reflected by the signal obtained from LR neurons. We therefore tuned $\tau$ so that the steady state reelin concentration in $Re^+$alECLII neurons was six times higher (528 $nM$), giving $\tau$ = 79000 $molecules \cdot h^{-1}$.

**Eq (3) parameters.** Assuming the intracellular half-life of Aβ$_{reelin}$ to be somewhat longer than that of Aβ and free reelin, we let it be 35 hours, giving $\delta$ = 0.02 $h^{-1}$.

**Eq (4) parameters.** Due to the lack of specific data, we assumed that the half-life of the reelin-induced phosphorylated state of GSK3β is approximately one hour, giving $\kappa$ = 0.65 $h^{-1}$. The estimated $K_d$ value of reelin binding to its receptors is approximately 0.5 $nM$ [9, 55], and to ensure a maximum effect of reelin signaling, the concentration of reelin must be several times higher [56]. Since experiments with embryonic brain cells derived from homozygous *RELN* mice suggest that the addition of 350-700 $pM$ reelin to the medium can induce substantial phosphorylation of GSK3β [57], this indicates that the surface density of ApoER2 in LR neurons is moderate. Due to this, we assumed it to be 36 $molecules \cdot \mu M^{-2}$, which implies $G_{tot}$ ([GSK3$_{tot}$]) = 50000. If $Re^+$alECLII neurons had the same value $G_{tot}$, it would mean that the

**Table 1. Parameter values used to describe LR and Re$^+$alECLII neurons.** Note that the $\alpha_{infection}$ value for Re$^+$alECLII neurons was found by demanding that the maximum [$A\beta$] level during infection should be about the same as for LR neurons. See main text for further explanation.

|  | LR neurons | Re$^+$alECLII neurons | Units |
|---|---|---|---|
| $\alpha_{baseline}$ | 1560 | 7020 | $molecules \cdot h^{-1}$ |
| $\alpha_{infection}$ | 44000 | 109000 | $molecules \cdot h^{-1}$ |
| $\beta$ | 0.1 | 0.1 | $h^{-1}$ |
| $\gamma$ | 0.005 | 0.005 | $molecules^{-1} \cdot h^{-1}$ |
| $\tau$ | 14500 | 79000 | $molecules \cdot h^{-1}$ |
| $\rho$ | 0.05 | 0.05 | $h^{-1}$ |
| $\delta$ | 0.02 | 0.02 | $h^{-1}$ |
| $\eta$ | 12 | 12 | $h^{-1}$ |
| $G_{tot}$ | 50000 | 250000 | $molecules$ |
| $\kappa$ | 0.65 | 0.65 | $h^{-1}$ |

response time would be considerably higher than that of LR neurons in terms of how fast the cell can ramp up its rate of p-tau production associated with an increase in the rate of A$\beta$ production. Therefore, we would expect that $G_{tot}$ is higher in these neurons. In fact, this seems to be the case [58], and we consequently let $G_{tot}$ = 250000, giving an ApoER2 surface density of 180 $molecules \cdot \mu M^{-2}$. This density is still quite moderate compared to the density of several surface receptors [59–61]. We let $\eta$ = 12 $h^{-1}$ to obtain a baseline proportion of phosphorylated GSK3$\beta$ of approximately 99% in both groups of neurons.

For convenience, all the parameter values above are compiled in Table 1.

## Software used

The model was coded in Python in a Jupyter lab environment. The SymPy Python library [62] was used to calculate closed-form expressions for the steady states of the four state variables under the baseline condition with a low and constant $\alpha$ value. The SciPy Python library [63] was used to study the time evolution of the differential equation system (Eqs (1)–(4)) (scipy. integrate.solve_ivp with the Radau solver).

## Results and discussion

### Mimicking an immune response in the two groups of neurons

Assuming that A$\beta$ is an important player in an intraneuronal immune response against pathogen infection under native physiological conditions, and based on what we now know about the relationship between reelin and A$\beta$ and the relationship between reelin and p-tau formation, we believe that the following scenario is likely to apply in most cortical neurons:

The production of A$\beta$ is very low as long as the cell does not detect any infection and there will be a marginal accumulation of A$\beta_{reelin}$ complexes. An infection will induce an abrupt increase in the production of A$\beta$. This will lead to increased accumulation of A$\beta_{reelin}$, depletion of the free reelin level, increase of the free A$\beta$ level, and marked formation of p-tau. After eliminating the pathogen, the cell reduces A$\beta$ levels back to baseline, allowing for a restoration of the original levels of free reelin and A$\beta_{reelin}$ and prevention of further p-tau formation through the reelin-GSK3$\beta$ pathway.

The *RELN* promoter appears to be under epigenetic control [64], but we assume that this is not relevant in our case. And we assume that reelin production is not under autoregulatory feedback control such that binding to A$\beta$ is not homeostatically compensated for by enhanced

reelin production. The reason being that binding between Aβ and reelin is likely to have a distinct physiological function in native physiology by allowing activation of the GSK3β pathway.

To assess how well the dynamic model was capable of recapitulating the above scenario in cortical neurons with high or low levels of constitutively produced reelin, we initially let a neuron produce Aβ ($\alpha(t)$) at a constant low rate, as described. The initial values of the four state variables at time $t_0$ were set identical to the steady state values under the given parameter regime (Eqs (5)–(8)). At an arbitrary time $t_1$ (50 hours), we assumed that the cell detects a pathogen invasion and immediately increases its Aβ production rate to a constant level several times higher. Except for presuming its existence, we were agnostic about the details of the sensing mechanism and how it induces an increase in Aβ production. At time $t_2$ (170 hours), we assumed that the cell detects that the pathogen has been eliminated and let the production rate of Aβ immediately return to the low basal level. We assumed that the infection lasted five days. However, this figure is not decisive, as a shorter or longer duration will only affect the size of the accumulated Aβ$_{reelin}$ pool and therefore the time it will take before the pool size returns to normal levels after the cessation of infection.

Since our main objective was to identify the prerequisites for achieving a sensible and similar immune response in both types of neurons under the assumption that the basic regulatory architecture is the same, this very simple way of describing a hypothetical immune response is arguably fit for purpose.

**Predicted immune response in LR neurons.**   To mimic the onset of the immune response in LR neurons after 50 hours, we increased the baseline Aβ production rate to $\alpha_{infection}$ = 44000 *molecules* · $h^{-1}$ to obtain an assumed Aβ steady state concentration of 100 *nM*. By this procedure, the model produced an immune response pattern (Fig 2) fully in line with the scenario described above. It should be emphasized that this pattern is obtained even when we let $\alpha_{infection}$ = 15000 *molecules* · $h^{-1}$, giving a Aβ steady state concentration of 1.8 *nM*. However, the Aβ concentration level during infection is likely to be much higher than this [65]. In any case, this shows that the immune response pattern shown in Fig 2 can be achieved by a wide range of $\alpha_{infection}$ values, implying that the current lack of experimental measurements of this parameter is not critical.

**Predicted immune response in Re⁺alECLII neurons.**   In Re⁺alECLII neurons, we had to increase the Aβ production rate to $\alpha_{infection}$ = 109000 *molecules* · $h^{-1}$ to obtain a Aβ steady state concentration of 100 *nM* (Fig 3). Comparing Figs 2 and 3, we see that the main difference is that the predicted level of Aβ$_{reelin}$ is more than five times higher in Re⁺alECLII neurons than in LR neurons. If cortical neurons possess an intraneuronal immune response against pathogens, the model suggests that a prerequisite for Re⁺alECLII neurons to produce an immune response on par with cells that have much lower constitutive levels of reelin is that they have a production rate of Aβ that is approximately 2.5 times higher during infection. For both groups of neurons, the model predicts that during infection there will be a conspicuous drop in the amount of free reelin and a pronounced increase in the amounts of Aβ and Aβ$_{reelin}$ (Figs 2 and 3).

## Validation of the model

We believe that all predictions provided by Figs 2 and 3 can be conclusively tested by infecting wild type rodents with, for example, herpes simplex virus 1 (HSV1) [26] and measuring the levels of free reelin, free Aβ and Aβ$_{reelin}$ before, during and after infection by existing immunohistochemical protocols [8]. Unfortunately, to the best of our knowledge, such data are not yet available. In the following, we provide a preliminary validation by two different approaches. The first shows that the model is capable of recapitulating observations associated with two contrasting genetic backgrounds, one preventing and one promoting AD development. The

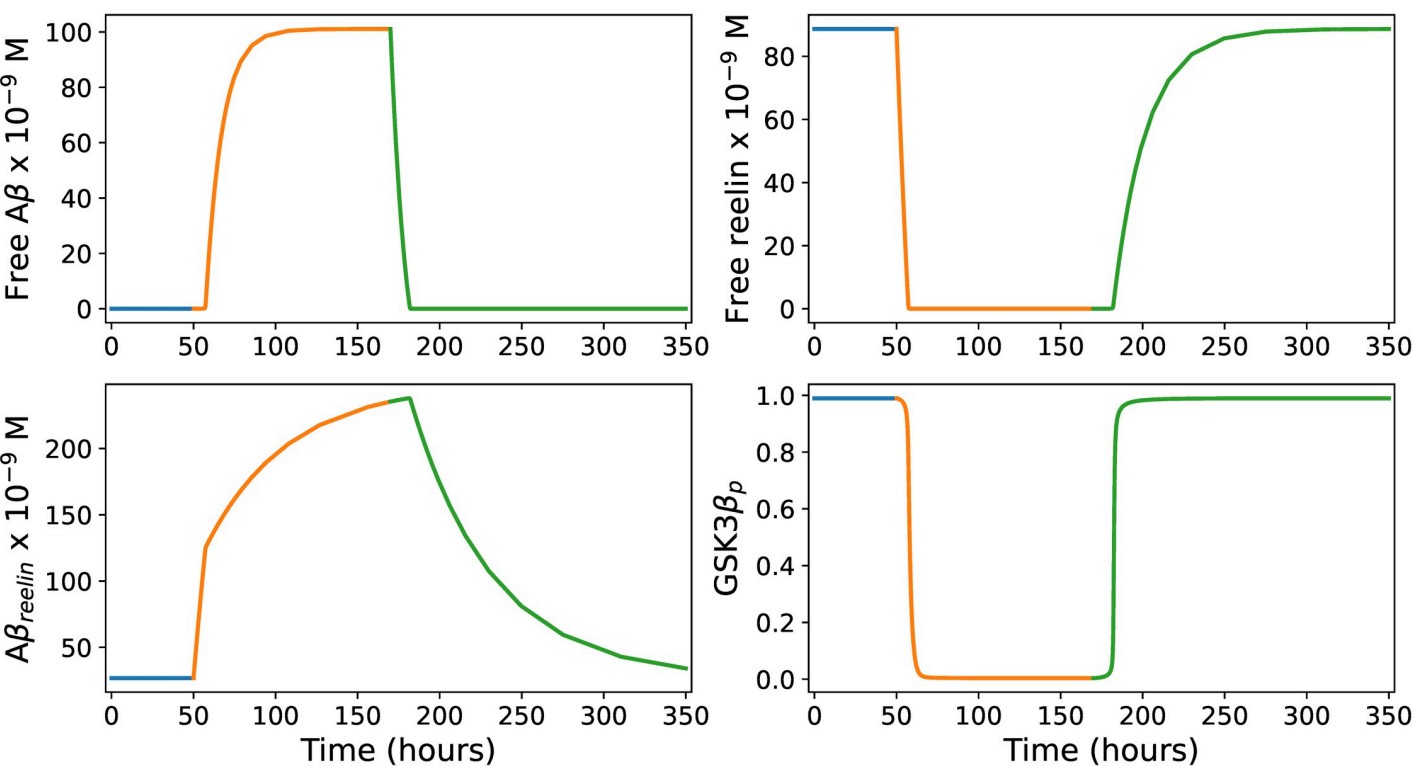

**Fig 2. Immune response in LR neurons.** The four panels show how the levels of free Aβ, free reelin, Aβ$_{reelin}$ and the proportion of phosphorylated GSK3β relative to the total amount of GSK3β ($G_{tot}$) develop during an infection event lasting 120 hours (from t = 50 to t = 170). The blue, orange, and green colors describe the pre-infection, infection, and post-infection phases, respectively. See the main text for further explanation.

second shows that the model can predict recent experimental data obtained by lowering the level of reelin in Re⁺alECLII neurons in McGill-R-Thy1-APP rats.

**The effect of genotype on the immune response in Re⁺alECLII neurons.** The recently discovered 'COLBOS' genotype, which refers to a rare mutation in the *RELN* gene that encodes reelin, has been shown to cause extreme resilience to AD even when the carrier also possesses the autosomal dominant familial AD (FAD) PSEN1-E280A mutation [66]. The *RELN* mutation clearly helps EC neurons maintain a level of functional reelin that is sufficient to prevent the formation of p-tau, despite being exposed to a high Aβ load [66]. One possible explanation is that the mutation causes a dramatic reduction in the affinity between Aβ and reelin. Thus, letting the value of the parameter γ (Eqs (1), (2) and (3)) become much less than that used to produce Figs 2 and 3, while keeping all other parameters fixed and not including the effect of the PSEN1-E280A mutation, the model can be used to predict the effect of this mutation.

As above, we allowed Re⁺alECLII neurons to remain at baseline steady state levels for 50 hours before changing the Aβ production rate from its base level to its maximum level for 120 hours. The model predicts that the level of free Aβ will increase approximately twofold compared to ApoEε3/ε3 Re⁺alECLII neurons, and that the Aβ$_{reelin}$ level will be approximately halved (Fig 4). However, the most notable prediction is that GSK3β will practically remain inhibited during infection, preventing p-tau formation. This is in complete agreement with the notion that the *RELN* mutation prevents p-tau formation by reducing the affinity between reelin and Aβ, causing the level of free reelin to stay above the threshold necessary to inhibit the activity of GSK3β kinase even when the neuron is exposed to a severe free Aβ burden [66].

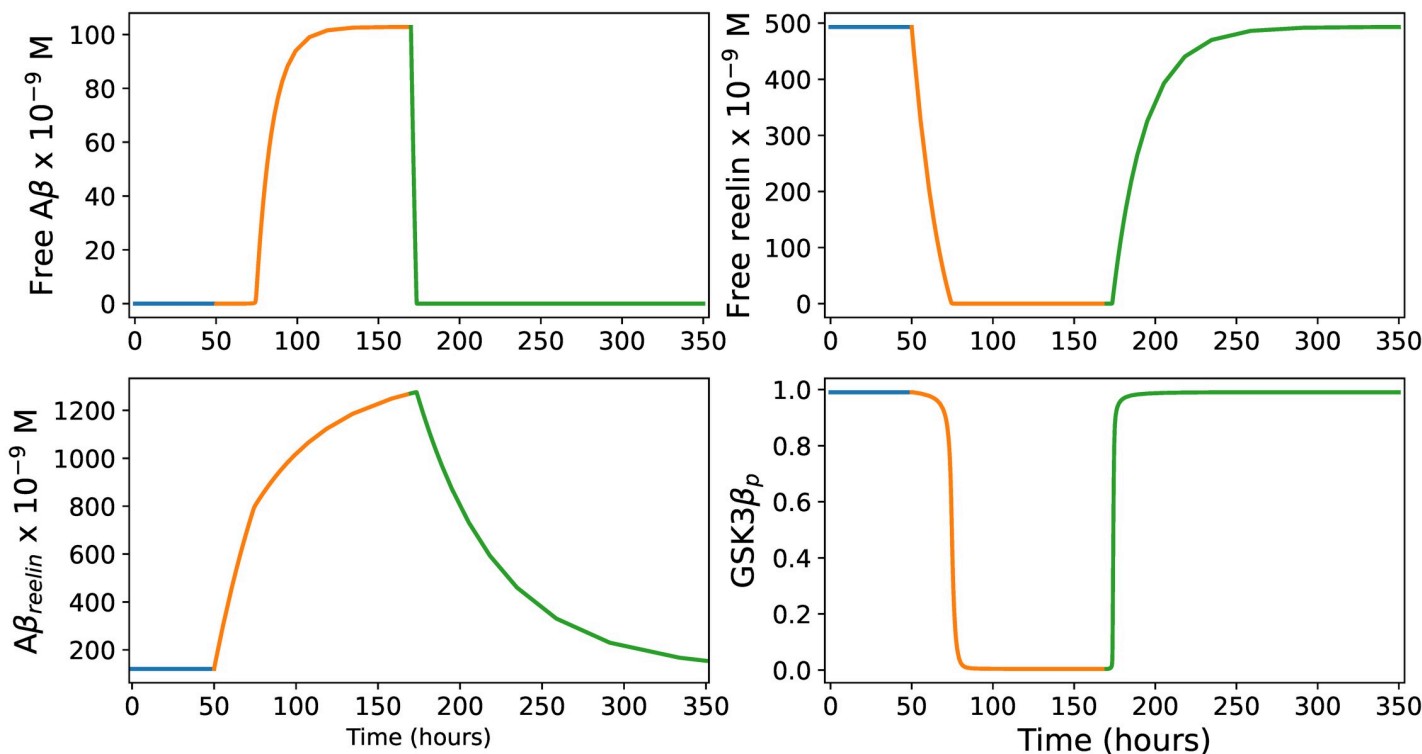

**Fig 3. Immune response in Re⁺alECLII neurons.** The four panels show how the levels of free Aβ, free reelin, Aβ$_{reelin}$ and the proportion of phosphorylated GSK3β relative to the total amount of GSK3β ($G_{tot}$) develop during an infection event lasting 120 hours (from t = 50 to t = 170). See the main text for further explanation.

On the opposite side of the COLBOS genotype in the AD risk spectrum, we have the ApoEε4/ε4 genotype. The propensity to develop sporadic AD is at least eight times higher in this genetic background than in the most common ApoE genotypic background (ApoEε3/ε3) [67]. This makes the ApoEε4/ε4 genotype the most pronounced genetic risk factor for the acquisition of sporadic AD. The causal basis for the observed difference between ApoEε4/ε4 and ApoEε3/ε3 subjects has become a much debated issue, and there are numerous attempts at explanations in the literature [68]. Reelin signaling involves endocytosis of ApoER2 receptors. ApoE isoforms differentially affect the recycling of these endosomes, where ApoEε3-containing endosomes readily recycle back to the surface, while those containing ApoEε4 tend to stall in the vesicles [69]. Here, we assume that this vesicle stalling causes some reelin to be removed from the circulation. Due to the lack of data on cycling rates, we simply mimicked this process by increasing the decay constant $\rho$ to 0.2 in Eq (2).

In the ApoEε4/ε4 background, the model predicts that the baseline steady state level of free reelin in Re⁺alECLII neurons will be markedly reduced (Fig 5). However, the most conspicuous prediction is that GSK3β phosphorylation will be less effective under baseline conditions. Given that the effect of vesicle stalling is sufficiently large, this implies that subjects carrying this genotype will have a higher constitutive production of p-tau that over years may have a substantial pathophysiological impact. It remains to be determined how important this effect is compared to other functional effects of the ApoEε4/ε4 genotype. However, it suggests that the vesicle stalling caused by this genotype may be of importance. And, considering that removal of tau reduces the formation of amyloid plaques in mouse models [70], our results are

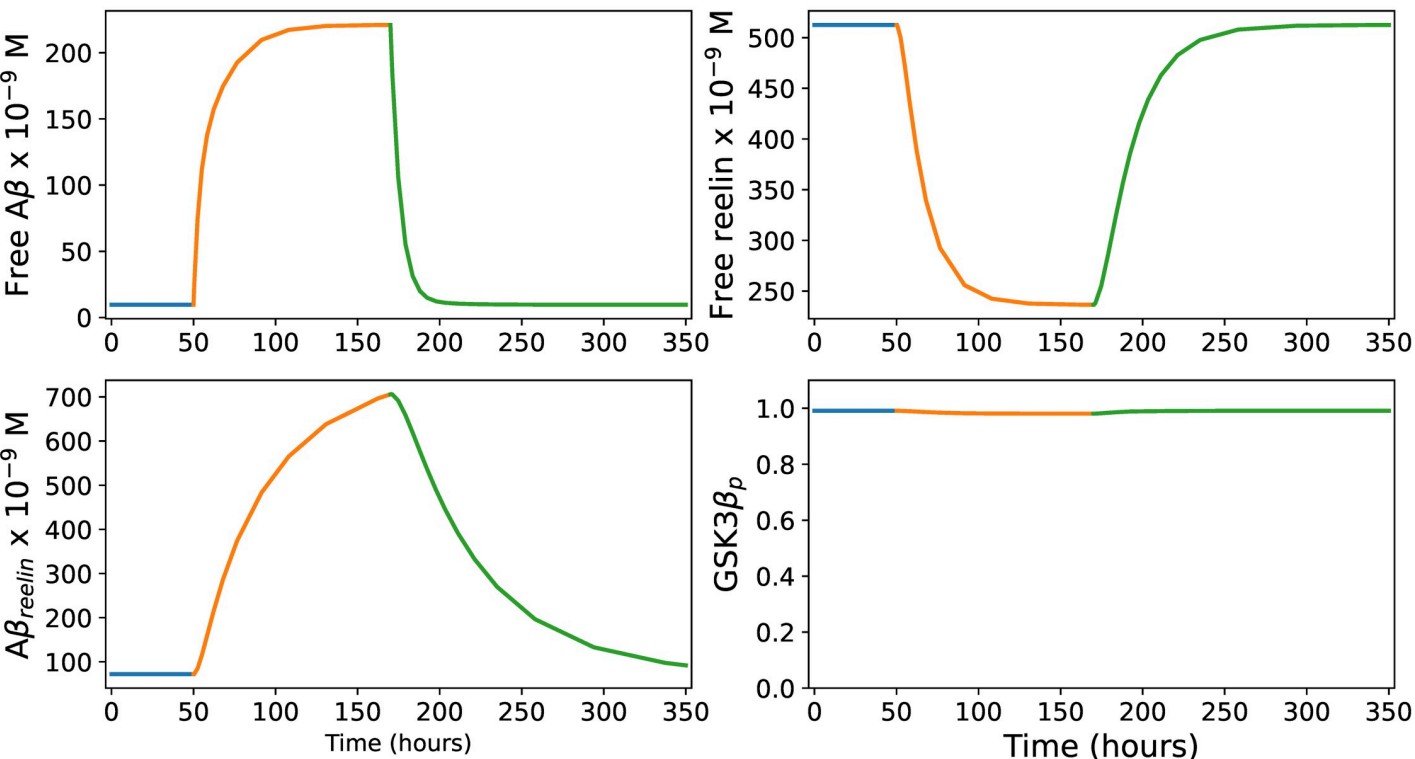

**Fig 4. Immune response in Re⁺alECLII neurons with the COLBOS reelin mutation.** The parameter values are identical with those of the ApoEϵ3/ϵ3 genotype, except that $\gamma$=0.0000001, i.e., we assume that the reelin mutation is genetically dominant and causes a dramatic reduction in affinity between reelin and Aβ. The four panels show how the levels of free Aβ, free reelin, Aβ_{reelin} and the proportion of phosphorylated GSK3β develop during an infection event lasting 120 hours (from t = 50 to t = 170). See the main text for further explanation.

consistent with the fact that ApoEϵ4/ϵ4 genotypes are characterized by an extremely elevated amyloid plaque burden [71].

Much as the model is capable of recapitulating key features associated with these two genetic backgrounds, which arguably to some degree validates the model as such, much stronger experimental confirmation of the model predictions is of course needed. Conditional on the availability of human neuronal cell cultures in which neurons express a phenotype very similar to Re⁺alECLII neurons [72], the above predictions can be tested in a setting where cells carrying ApoEϵ4/ϵ4, ApoEϵ3/ϵ3 and COLBOS genotypes are temporarily exposed to a substantially increased intracellular level of Aβ.

**Predicting the effect of lowering the level of reelin in Re⁺alECLII neurons.** The test data (Fig 6A and 6B) were taken from a recent experimental study in McGill-R-Thy1-APP homozygous transgenic rats [8]. These rats carry a transgene containing human APP751 with Swedish double mutations and Indiana mutations expressed under the control of the murine Thy1.2 promoter. The transgene causes an increase in the basal production rate of Aβ and an increase in the fraction of Aβ42. Here we just briefly sketch how the test data were generated, since all experimental protocols are given in reference [8].

The reelin level was reduced by injecting Re⁺alECLII neurons with an adeno-associated viral vector (AAV) expressing miRNA against reelin into the left or right alEC of five one month old McGill-R-Thy1-APP rats. For each injection, the control vector was injected into the contralateral alEC. Reelin levels and Aβ42 levels in experimental neurons (N = 820) and

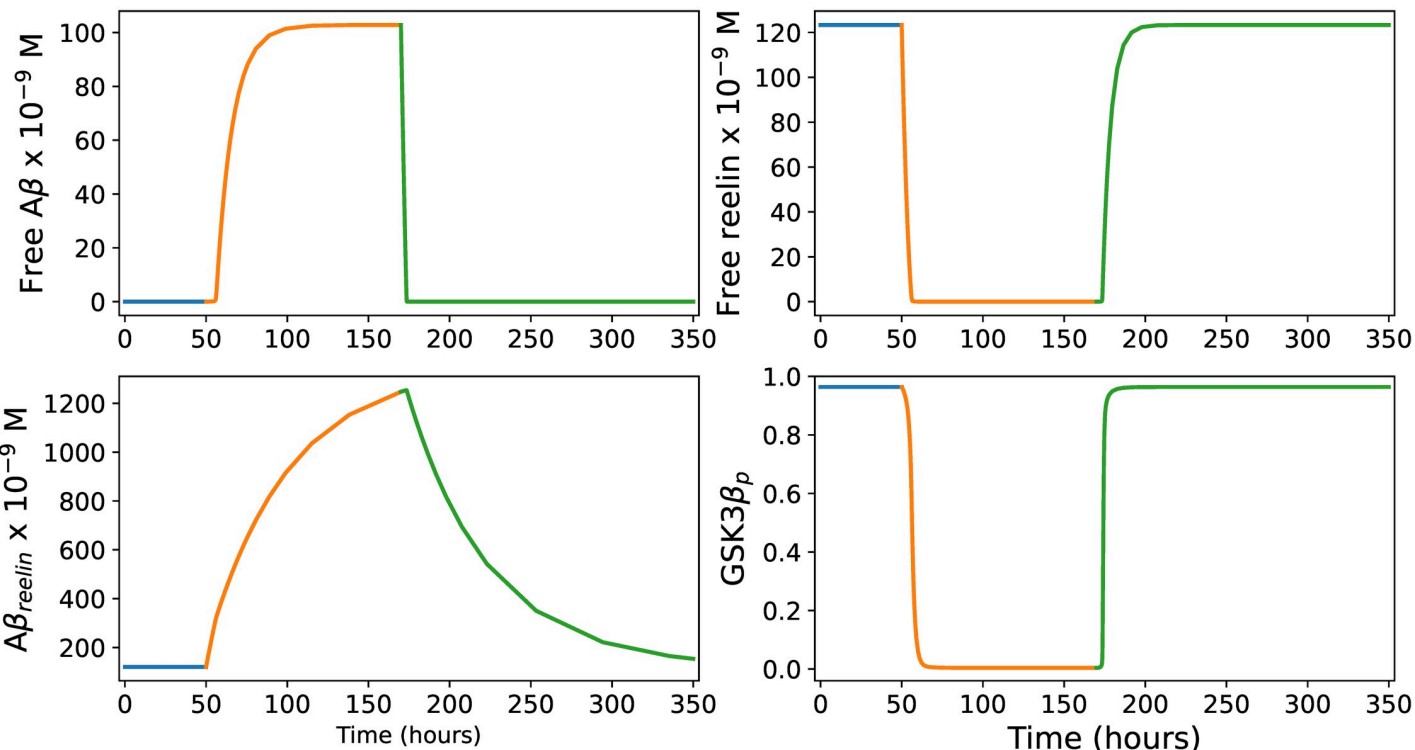

**Fig 5. Immune response in ApoEε4/ε4 Re⁺alECLII neurons.** The parameter values for the ApoEε4/ε4 genotype are identical with those of the ApoEε3/ε3 genotype, except that $\rho$=0.2. The four panels show how the levels of free Aβ, free reelin, Aβ$_{reelin}$ and the proportion of phosphorylated GSK3β develop during an infection event lasting 120 hours (from t = 50 to t = 170). See the main text for further explanation.

control neurons (N = 585) were quantified by fluorescence densitometry when the animals were two months old. The mean levels of reelin and of Aβ42 were in the experimental group of neurons measured as 26% (Fig 6A) and 69% (Fig 6B) of the mean levels in the control group, respectively.

To test whether the model could generate the above two percentage figures, we kept all parameter values used in the wild type Re⁺alECLII neuron case (Table 1) unchanged except for the baseline production rate of Aβ42 ($\alpha_{baseline}$) and the production rate of reelin ($\tau$). The value of $\alpha_{baseline}$ was changed to account for the effect of the transgene in McGill-R-Thy1-APP rats, and the value of $\tau$ was changed to account for the effect of miRNA intervention. It should be noted that the test rests on the assumption that the given parameterization of Eqs 1, 2 and 3 also apply to rats. That is, we assume that the decay rates ($\beta, \rho, \delta$) and the binding rate between Aβ42 and reelin ($\gamma$) are about the same and that the ratio between the baseline production rates of Aβ42 ($\alpha$) and reelin ($\tau$) is approximately the same in humans and wild type rats.

We do not know exactly how much larger the baseline production rate of Aβ42 is in McGill-R-Thy1-APP Re⁺alECLII neurons relative to that of wild type. However, based on the expression level of the transgene [73] and existing reports that include measurements of Aβ42 [74, 75], we can confidently assume that it will be in the range of 2-10 times that of wild type alECLII neurons (Table 1). The miRNA-driven reduction of the reelin mRNA pool can be directly translated into a reduced reelin production rate ($\tau$), and we allowed $\tau$ to vary in the range 5-100% of the wild type value.

In the control neuron case, we then calculated an array of possible steady states of the total copy number of reelin (using Eq (6) + Eq (7)) and the total copy number of Aβ42 (using Eq (5)

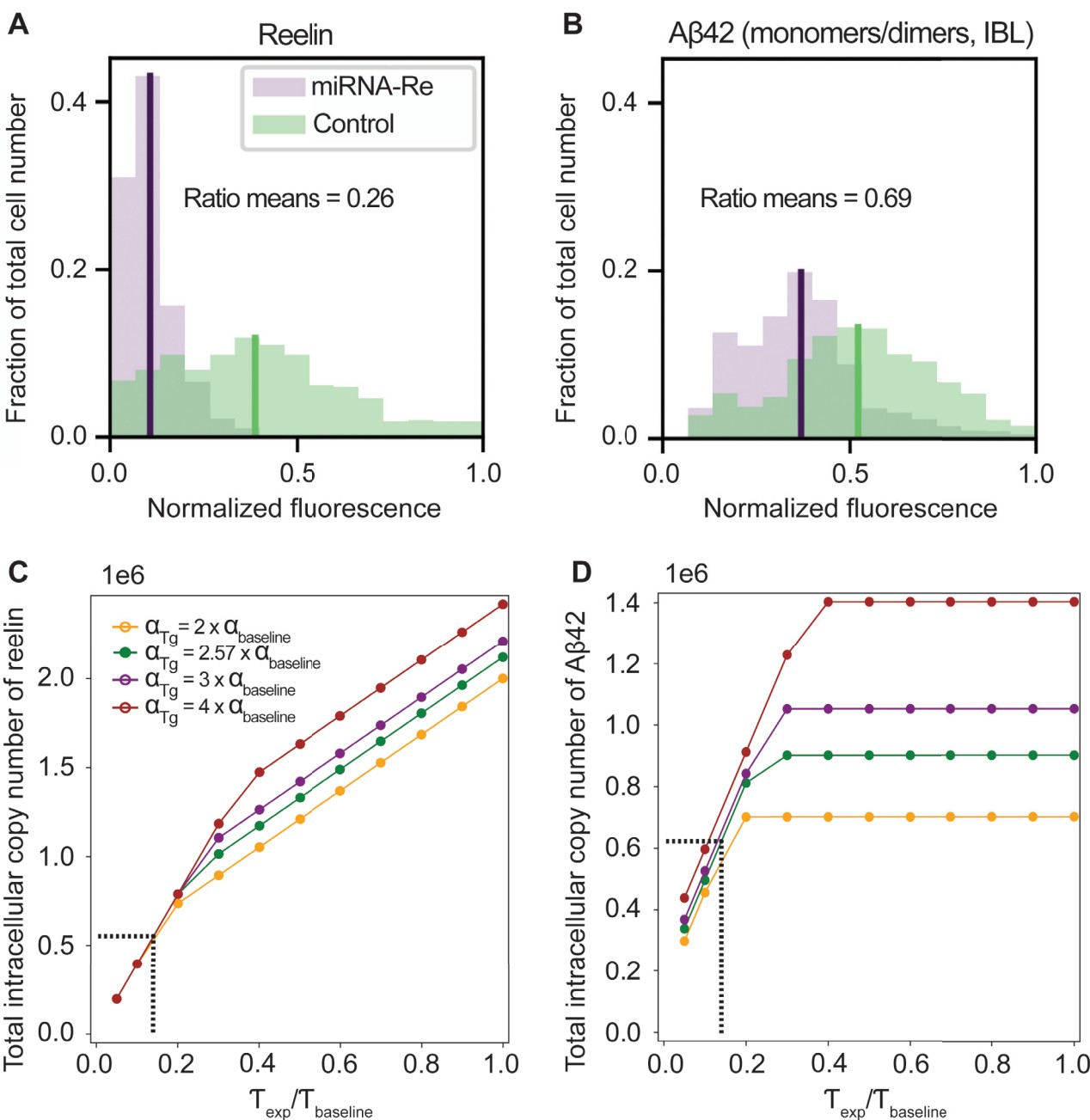

**Fig 6. Recapitulating the effects of miRNA intervention in Re⁺alECLII neurons of McGill-R-Thy1-APP rats.** Panels A and B are identical with panels 1 and 4 in Fig 4A in reference [8]. The black vertical lines denote the mean values of the distributions in the experimental group of neurons. The green vertical lines denote the mean values of the distributions in the control group of neurons. The ratios of the mean values of the experimental group to the mean values of the control group are 0.26 and 0.69, respectively. C. Predicted steady state total number of reelin in experimental neurons as a function of the fold change of $\alpha_{baseline}$ in transgenic rats (Tg) compared to wild type and the fractional decrease of $\tau$ relative to wild type. Only a small subset of the fold change values and the fractional change values used in the numerical study is shown. The vertical dashed line denotes the predicted fractional reduction in the reelin production rate relative to wild type (0.14) that best fits the data. The horizontal dashed line denotes the total copy number of reelin when the fold change is 2.57. D. Predicted steady state total copy number of Aβ42 in experimental neurons as a function of the fold change of $\alpha_{baseline}$ relative to wild type and the fractional decrease of $\tau$ relative to wild type. The horizontal dashed line denotes the total copy number of Aβ42 when the fold change is 2.57.

+ Eq (7)) by letting the fold change of $\alpha_{baseline}$ relative to wild type vary in steps of 0.005 from 2 to 10. In the experimental neuron case, we let $\alpha_{baseline}$ vary in the same way as above. In addition, for each value of $\alpha_{baseline}$, we let the fractional change in $\tau$ vary in steps of 0.005 from 0.05 to 1.0. For each step, we calculated the fraction of the total copy number of reelin and Aβ42 in the experimental group relative to the control group. This gave us 304380 records, each containing the fold change in $\alpha_{baseline}$ relative to wild type, the fractional change in $\tau$ relative to wild type, and the associated fractions of total reelin and total Aβ42. In this set, we then searched for the record that gave the closest fit to the data.

We found that the model can exactly recapitulate the observed mean fractional decrease in total reelin (0.26) and total Aβ42 (0.69) when the value of $\tau$ in the experimental neuron is 14% of the value of the control neuron and $\alpha_{baseline}$ is 2.57 times the value of the wild type (Fig 6C and 6D).

It should be noted that the experimental data only reflect the amount of monomeric and dimeric Aβ42, and we do not know how much was present in the prefibrils or protofibrils. However, we have no apparent reason to believe that the fractions of the three groups differed very much between experimental neurons and control neurons [8] and therefore the reported ratio of monomeric and dimeric Aβ42 (0.69) is likely to be representative of the total amount of Aβ42.

Considering the strong left skewness of the miRNA-Re distribution (Fig 6A) and that its mean value is approximately 12% of the maximal fluorescence level measured, the predicted miRNA-driven reduction in the effective rate of reelin production (14%) appears sensible. To our knowledge, experimental data conclusively supporting or refuting the predicted fold change (2.57) in $\alpha_{baseline}$ are not yet available. However, it is consistent with existing data [73, 74].

The predicted values depend on parameterization. To probe this dependency, we reduced the wild type value of $\tau$ in Re$^+$alECLII neurons (Table 1) with 50%. We then repeated the numerical experiment in McGill-R-Thy1-APP rats described above. In this case, the fold change in $\alpha_{baseline}$ must be 1.28 to get the ratio values 0.26 and 0.69. Since this clearly discords with data on the effect of the transgene in McGill-R-Thy1-APP rats [73, 74], it indicates that the value of $\tau$ used to produce Fig 2 is not completely off the mark. Due to the current lack of critical experimental data that can be used to further constrain the model parameters, it is nevertheless premature to perform an exhaustive analysis of the relationship between parameter values and the degree to which the model can account for the experimental data in reference [8]. However, the above analysis suggests that this is a worthwhile undertaking to improve the realism of the model as soon as we have better quantitative experimental data, in particular data on the Aβ42 dynamics in Re$^+$alECLII neurons of McGill-R-Thy1-APP and wild type rats.

Taken together, the above results suggest that we are positioned to claim that the model is capable of making predictions that are in clear agreement with the miRNA results on Re$^+$alECLII neurons [8]. This implies that it adequately captures the steady state relationships between Aβ42 and reelin, strongly supporting its main premise, namely that reelin functions as a sink for Aβ42. This, of course, does not serve as a satisfactory validation of the predicted dynamic behavior of the model (Figs 1 and 2). We have alluded to the type of experiments needed to test this behavior. Considering the deliberate simplicity of the model and the uncertainty attached to most of the parameters, it would be surprising if it turned out that new quantitative test data would not require revision of parameter values and adjustments of the model. However, we anticipate that such revisions will not affect our main conclusions.

## Concluding remarks

Although it has been established that Aβ has antiviral and antimicrobial properties, it remains to be conclusively shown that Aβ is indeed a major player in a concerted intraneuronal

immune response. But if there is such a response, we claim that $Re^+alECLII$ neurons will have to sport a much higher maximum production capacity for Aβ than LR neurons due to their extraordinarily high reelin production. This enhanced production capacity resolves Paradox 3 stated in the Introduction by the fact that it allows $Re^+alECLII$ neurons to exert an immune response on the same level as LR neurons. In a senescent physiology predisposing to frequent inflammation-driven Aβ production bursts, the model predicts that $Re^+alECLII$ neurons will repeatedly produce enough Aβ to cause the amount of free reelin to drop so much that the formation of p-tau is initiated. Thus, as alluded to, the resolution of Paradox 3 appears to also resolve Paradoxes 1 and 2 stated in the Introduction.

Here, we have deliberately focused on studying the operation of the assumed immune response in native physiology using a very simplistic model. Modeling the effects of a dysfunctional operation of this immune response in a senescent physiology characterized by recurrent inflammation events is a much more challenging undertaking that is beyond the scope of this study. Specifically, this would have to include descriptions of the fate of Aβ, $Aβ_{reelin}$ and p-tau tagged for degradation by the endosome-autophagosome-lysosome system. The disruption of this system consistently appears as an early and progressive characteristic of the pathophysiology of AD [76], leading to the accumulation of degradation-tagged cellular cargo. A solid body of data implicates that the dysfunction of the lysosomal vacuolar (H+)–adenosine triphosphatase (v-ATPase) proton pump is of particular significance and that the amyloid precursor protein (APP) is instrumental in the alkalinization of lysosomes and the subsequent decrease in the lysosomal degradation rate [77–80].

Based on the empirical data available, it is therefore fully conceivable that the degradation of Aβ and $Aβ_{reelin}$ during an inflammation event follows zero-order kinetics instead of first-order kinetics due to the APP-driven reduction of lysosomal degradation capacity. That is, the degradation machinery becomes rapidly saturated, causing a marked increase in the time required to return to baseline levels. This would imply that if the recurrent inflammation events occur frequently and/or last for a prolonged time, the levels of Aβ and $Aβ_{reelin}$ can over years reach pathophysiological levels in terms of soma volume occupancy and subsequent detrimental effects on a variety of cellular processes. This scenario is particularly relevant for the $Aβ_{reelin}$ complex because of the large size of the reelin molecule. This reasoning is supported by experimental data showing that intracellular accumulations of $Aβ42$ positive material ranged from 20 to 80% of the total cytoplasmic volume in neurons of the entorhinal cortex of patients with sporadic AD [49, 50]. And, as advocated by an increasing number of researchers [48, 49, 81], it points to the possibility that plaque formation can also occur intraneuronally. Thus, given the recurrent dysfunctional activation of the native immune response mechanism studied here, this suggests that the associated long-term effects involve much more than just p-tau seeding. Therefore, we anticipate that if our model is adapted to a senescent physiological context, it may provide crucial information about why ECLII neurons are a cradle for AD.

## Acknowledgments

We are very grateful to Sir Peter Hunter and David Nickerson for translating the model into CellML code and making it available on the Physiome Model Repository.

## Author Contributions

**Conceptualization:** Asgeir Kobro-Flatmoen, Stig W. Omholt.

**Formal analysis:** Asgeir Kobro-Flatmoen, Stig W. Omholt.

**Funding acquisition:** Asgeir Kobro-Flatmoen.

**Methodology:** Asgeir Kobro-Flatmoen, Stig W. Omholt.

**Project administration:** Asgeir Kobro-Flatmoen.

**Writing – original draft:** Asgeir Kobro-Flatmoen, Stig W. Omholt.

**Writing – review & editing:** Asgeir Kobro-Flatmoen, Stig W. Omholt.

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
