## [Decision Letter · Decision Letter 0]

13 Sep 2024

Dear Dr. Kobro-Flatmoen,

Thank you very much for submitting your manuscript "Intraneuronal binding of amyloid beta with reelin - implications for the onset of Alzheimer’s disease" for consideration at PLOS Computational Biology.

As with all papers reviewed by the journal, your manuscript was reviewed by members of the editorial board and by several independent reviewers. In light of the reviews (below this email), we would like to invite the resubmission of a significantly-revised version that takes into account the reviewers' comments.

We cannot make any decision about publication until we have seen the revised manuscript and your response to the reviewers' comments. Your revised manuscript is also likely to be sent to reviewers for further evaluation.

Sincerely,

Fabian Spill

Academic Editor

PLOS Computational Biology

Jason Haugh

Section Editor

PLOS Computational Biology

Reviewer's Responses to Questions

**Comments to the Authors:**

Reviewer #1: review attached

Reviewer #2: This work tried to represent protein-protein interaction based on Alzheimer's disease phenomenon. The authors highlighted problems clearly and it was really engaging because they explained paradoxical hypotheses. The mathematical modelling could help to resolve the problems. However, I have some notes to be considered:

1. Could you please add some mathematical stability analysis for the obtained equilibrium and show other possible steady state equilibrium points (if any)? This additional analysis can be included as supplementary information. This could be important for other research fellows who work with mathematical modelling.

2. Could you please add some data or data analysis to support your results if any? It is not only to validate your model, but also to trigger further experiments because in your work you tried to resolve paradoxical hypotheses.

**Have the authors made all data and (if applicable) computational code underlying the findings in their manuscript fully available?**

Reviewer #1: **No: **i can't see the code

Reviewer #2: **No: **The authors only provided information about coding library used for their work. However, the authors didn't make the code available online.

PLOS authors have the option to publish the peer review history of their article (what does this mean?). If published, this will include your full peer review and any attached files.

Reviewer #1: No

Reviewer #2: No
---

## [Decision Letter · Decision Letter 1]

9 Dec 2024

Dear Dr. Kobro-Flatmoen,

We are pleased to inform you that your manuscript 'Intraneuronal binding of amyloid beta with reelin - implications for the onset of Alzheimer’s disease' has been provisionally accepted for publication in PLOS Computational Biology.

One typo thas was flagged by a reviewer was as follows:

' the punctuation around ’COLBOS’ on line 394 is incorrect. '

Best regards,

Fabian Spill

Academic Editor

PLOS Computational Biology

Jason Haugh

Section Editor

PLOS Computational Biology

Reviewer's Responses to Questions

**Comments to the Authors:**

Reviewer #1: I think this paper is a great addition to the field and more studies like this are needed to gain a deeper understanding of this complex disease and its driving mechanisms. I appreciate your thoughtful responses to my feedback and the careful incorporation of my suggestions.

Reviewer #2: Thank you for the response. I would consider it is now completed.

**Have the authors made all data and (if applicable) computational code underlying the findings in their manuscript fully available?**

Reviewer #1: Yes

Reviewer #2: Yes

PLOS authors have the option to publish the peer review history of their article (what does this mean?). If published, this will include your full peer review and any attached files.

Reviewer #1: No

Reviewer #2: No

---

## [Editor Report · Acceptance letter]

31 Dec 2024

PCOMPBIOL-D-24-01017R1 

Intraneuronal binding of amyloid beta with reelin - implications for the onset of Alzheimer’s disease

Dear Dr Kobro-Flatmoen,

I am pleased to inform you that your manuscript has been formally accepted for publication in PLOS Computational Biology. Your manuscript is now with our production department and you will be notified of the publication date in due course.

With kind regards,

Anita Estes
